# The Role of the Paraventricular-Coerulear Network on the Programming of Hypertension by Prenatal Undernutrition

**DOI:** 10.3390/ijms231911965

**Published:** 2022-10-08

**Authors:** Bernardita Cayupe, Blanca Troncoso, Carlos Morgan, Patricio Sáez-Briones, Ramón Sotomayor-Zárate, Luis Constandil, Alejandro Hernández, Eugenia Morselli, Rafael Barra

**Affiliations:** 1Centro de Investigación Biomédica y Aplicada (CIBAP), Escuela de Medicina, Facultad de Ciencias Médicas, Universidad de Santiago de Chile (USACH), Santiago 9170020, Chile; 2Escuela de Enfermería, Facultad de Ciencias Médicas, Universidad de Santiago de Chile, Santiago 9170020, Chile; 3Laboratorio de Neurofarmacología y Comportamiento, Escuela de Medicina, Facultad de Ciencias Médicas, Universidad de Santiago de Chile, Santiago 9170020, Chile; 4Laboratorio de Neuroquímica y Neurofarmacología, Centro de Neurobiología y Fisiopatología Integrativa, Instituto de Fisiología, Facultad de Ciencias, Universidad de Valparaíso, Valparaíso 2360102, Chile; 5Laboratorio de Neurobiología, Departamento de Biología, Facultad de Química y Biología, Universidad de Santiago de Chile, Santiago 9170020, Chile; 6Department of Basic Sciences, Faculty of Medicine and Sciences, Universidad San Sebastián, Santiago 7510157, Chile

**Keywords:** prenatal undernutrition, hypertension, paraventicular nucleus, locus coeruleus, neural network

## Abstract

A crucial etiological component in fetal programming is early nutrition. Indeed, early undernutrition may cause a chronic increase in blood pressure and cardiovascular diseases, including stroke and heart failure. In this regard, current evidence has sustained several pathological mechanisms involving changes in central and peripheral targets. In the present review, we summarize the neuroendocrine and neuroplastic modifications that underlie maladaptive mechanisms related to chronic hypertension programming after early undernutrition. First, we analyzed the role of glucocorticoids on the mechanism of long-term programming of hypertension. Secondly, we discussed the pathological plastic changes at the paraventricular nucleus of the hypothalamus that contribute to the development of chronic hypertension in animal models of prenatal undernutrition, dissecting the neural network that reciprocally communicates this nucleus with the locus coeruleus. Finally, we propose an integrated and updated view of the main neuroendocrine and central circuital alterations that support the occurrence of chronic increases of blood pressure in prenatally undernourished animals.

## 1. Introduction

Since the early studies of David Barker and colleagues, much evidence has accumulated demonstrating that environmental conditions during early life are critical for the proper development of organisms, supporting the concepts of fetal programming [1] and developmental origin of health and disease (DOHAD) [2]. In this context, nutrition is one of the most relevant conditions that impacts early development generating deleterious long-term effects, thereby constituting a crucial pathognomonic component of chronic non-communicable diseases [3]. Caloric and protein malnutrition defined, respectively, as deficiencies in either total food or protein intakes, can occur as specific or mixed forms in the newborn and in later childhood [4], whereas malnutrition in pregnant mothers likely involves inadequate nutrition of the fetus both in calories and proteins. Such chronic fetal macronutrient deficiency manifests as full-term babies with low birth weight, i.e., small for gestational age (SGA). Although fetal undernutrition is not completely dependent on maternal nutrition (there also are other well-known determinants, like a poor maternal metabolic and endocrine status, abnormal feto-placental perfusion, smoking during pregnancy), the main factor determining SGA newborns in developing countries and second in importance in developed countries, is a low energy intake [5]. Undernutrition during fetal life have been shown to produce long-term adaptations in humans and animals, such as lower body weight and brain alterations, including loss of neurons and glia, impaired neuronal differentiation, and deficits in neuroplasticity [6,7]. In addition, several reports have shown that prenatal undernutrition produces metabolic disparities and cardiovascular illnesses, such as hypertension [3,8,9].

The Food and Agriculture Organization of the United Nations claims that millions of persons worldwide are born with low birth weight due to prenatal undernutrition [10,11], and many epidemiological studies have reported that prenatal undernutrition is associated with an increased risk of developing long-term diseases, like hypertension [12,13]. For instance, mothers exposed to the Dutch famine during gestation gave rise to offspring predisposed to develop hypertension in middle age [14] and, more recently, a higher incidence of chronic hypertension was reported in the Chinese population that suffered famine between 1959 and 1961 [15]. Indeed, few studies have quantified the effect of low birth weight on blood pressure in humans. Painter et al. [16] found that for every kilogram reduction in birth weight due to maternal exposure to famine there was a 2.7 mmHg increase in systolic pressure. More recent studies have established a significant increase in the prevalence of hypertension due to prenatal famine, the prevalence being 10.2% in the group without exposure to famine and 16.1% for those with fetal exposure to famine. These data were calculated from Chinese populations exposed to famine during 1959–1961 [17]. A fairly similar result was described in an Ethiopian population cohort composed of individuals exposed and not exposed to prenatal famine (1983–1985), where 64/350 (18.3%) of exposed individuals and 31/350 (8.9%) of unexposed individuals presented with hypertension. In that study, it was found that systolic blood pressure increased by 1.05 mmHg (95% CI 0.29, 4.42) and diastolic blood pressure by 2.47 mmHg (95% CI 1.01, 3.95), and multivariate logistic regression analysis of the model indicated a positive association between famine exposure and risk of hypertension in adults [18].

Low-birth-weight human patients also showed activation of the hypothalamic-pituitary-adrenal (HPA) axis along with high fasting cortisol levels when challenged with exogenous adrenocorticotropic hormone (ACTH), which are thought to be involved in the hypertension exhibited by these patients [19]. Similarly, persistent autonomic dysfunction is programmed in low-birth-weight human patients, as revealed by increased cardiac sympathetic activity at rest and reduced cardiac reflexes in response to head tilt [20]. In animal models, rats with prenatal and perinatal malnutrition showed chronic increases in systolic pressure and heart rate during adulthood [21,22,23,24,25], and an increased neuronal tone of the sympathetic peripheral system [26]. Therefore, protein and calorie food restrictions in pregnant dams result in hypertensive adult offspring [27,28] through mechanisms that include altered hypothalamic programming [19,26,29], sustaining the significant role of nutritional programming on the neural and endocrine control of the cardiovascular system. Indeed, an increase in the activity and excitability of neurons in the hypothalamic paraventricular nucleus (PVN) has been described in some animal models of hypertension, which are closely related to an increase in sympathetic impulse. For instance, electrophysiological studies in spontaneously hypertensive rats have shown higher neuronal activity of presympathetic PVN neurons that innervate the nucleus tractus solitarius [11,13,30,31]. Although GABAergic input normally inhibits presympathetic PVN neurons in normotensive rats [32,33], this effect is impaired in spontaneous hypertensive rats due to a post-translational modification (glycosylation) of the Na^+^-K^+^-2Cl^−^ cotransporter 1, resulting in an increased sympathetic tone [34]. Impairment of GABAergic inhibitory input in the dorsomedial and paraventricular hypothalamic nuclei has also been described in a model of neurogenic hypertension, the Schlager mouse [35,36]. In addition, epigenetic modulation of hypothalamic angiotensin signaling in the PVN has been reported to contribute to salt-sensitive hypertension induced by prenatal glucocorticoid excess in offspring of mothers that were undernourished during pregnancy [37].

The present review summarizes the primordial neuroendocrine and neuroplastic mechanisms that may explain the chronic increase in blood pressure in the offspring of prenatally malnourished individuals. Additionally, we discuss how the PVN and other brain nuclei are critical neuroadaptive components of the neural networks that may cause and maintain hypertension in these animals. Peripheral changes (e.g., renal, vascular) that may partly account for fetal undernutrition-induced hypertension in later life have been reviewed elsewhere and are not the aim of the present review (for review of renal changes, see [38,39]; for review of vascular adaptations, see [40,41,42]).

## 2. Hypothalamic CRF Expression and Subsequent Development of Hypertension in Subjects Who Suffered from Prenatal Undernutrition: The Role of Central Noradrenaline

Hypophysiotropic neurons synthesize the corticotropin releasing factor (CRF) in the medial parvocellular division of the hypothalamic PVN, which is then released from the median eminence, activating anterior pituitary corticotroph cells to release ACTH into the systemic circulation [43]. In turn, ACTH induces the synthesis and release of glucocorticoids (GC) in the cortex of the adrenal glands [44], which at basal levels maintain cardiovascular homeostasis by acting on the vasculature and the heart [45]. In addition to the hypophysiotropic cells that give rise to the HPA axis, there are other CRF-synthesizing neurons in the PVN, the preautonomic neurons, that project to autonomic centers in the brainstem and spinal cord, providing essential control over the responses of the cardiovascular system acting in parallel with HPA [46].

### 2.1. Dysregulation of Hypothalamic CRF Levels in Prenatally Undernourished Subjects: The Role of the Glucocorticoid Feedback and Placental Barrier

The HPA axis is highly sensitive to nutritional insults during prenatal life. Indeed, animals exposed to maternal undernutrition have higher CRF mRNA and protein levels in the hypothalamus [47] and ACTH plasma levels in adulthood [47,48,49]. Additionally, it has been shown that young children with low birth weight—an index of intrauterine growth restriction—have a higher risk of developing hypertension, together with higher serum levels of cortisol, ACTH, and CRF [50]. In addition, post-mortem studies in hypertensive patients have shown higher levels of hypothalamic CRF mRNA and protein [51]. Interestingly, an increase in spontaneous neuronal activity was observed in the PVN of adult rats previously exposed to prenatal undernutrition [52], suggesting that parvocellular neurons of the PVN in these animals not only express but also release more CRF because of the higher rate of neuronal firing. Thus, CRF-mediated hyperactivity of the HPA axis could lead to increased plasma corticosterone and higher blood pressure [53], whereas hypothalamic CRF could also trigger the activation of the sympathetic–adrenal–medullary axis [54], greatly contributing to the increase in blood pressure. Therefore, both the hypophyseal–adrenal axis and the sympathetic–adrenal–medullary system constitute output mechanisms whereby CRFergic neurons of the PVN would trigger hypertension in previously undernourished adult subjects. Through these mechanisms, PVN neurons can control peripheral cardiovascular viscera via corticosterone secretion and noradrenaline/adrenaline release, respectively.

It is therefore apparent that increased levels of hypothalamic CRF are in the origin of prenatal undernutrition-induced hypertension. Thus, what is the main factor underlying the increased hypothalamic CRF levels in adult hypertensive animals previously undernourished during intrauterine life? Published reports indicate that rodents exposed to prenatal undernutrition have a reduction in the expression of glucocorticoid receptors (GR) in the hippocampus [55], hypothalamus [56] and pituitary [57] at birth, adolescence, and adult life. These persistent decreases in GR expression found in previously undernourished animals could lead to reduced feedback control at the hypothalamus, thereby resulting in higher CRF levels together with hyperactivity of both the HPA axis and sympathetic output in adulthood. It is noteworthy that this deleterious effect on the hypothalamic glucocorticoid feedback caused by early undernutrition was confirmed with a glucocorticoid challenge functional test, involving injections of subcutaneous (100 μg/kg) and intra-PVN (50 pmol) dexamethasone in adult rats that had suffered from malnutrition during prenatal life. In that study, these treatments gave rise to reduced serum corticosterone levels in eutrophic but not in prenatally malnourished animals [58]. Furthermore, bilateral intra-PVN microinjection of GR agonists and antagonists showed significantly smaller effects on systolic pressure, heart rate, and plasma glucocorticoids when administered to prenatally malnourished rats, compared to eutrophic controls [28]. Therefore, a persistent reduction in GR expression in these brain areas could result in a decreased negative feedback control leading to increased activity of both the HPA axis and sympathetic output in adult life. This claim is endorsed by human and animal studies where higher expression levels of CRH, ACTH, and corticosterone/cortisol were observed in offspring of prenatally malnourished individuals [28,47,50]. One of the supporting mechanisms explaining reduced GR expression after prenatal undernutrition involves a malfunction at the placental barrier. Usually, maternal glucocorticoids should not cross the placental barrier. However, the placental activity of 11β-hydroxysteroid dehydrogenase type 2—the enzyme that catalyzes rapid metabolism of cortisol and corticosterone to inert steroids—is significantly decreased in fetuses of rats exposed to maternal malnutrition [19,55,56,59], thereby resulting in overexposure of the fetuses to maternal glucocorticoids. This overexposure phenomenon has also been reported in babies with reduced birth weight [60]. Finally, increased glucocorticoid signaling to the PVN would lead to down-regulation of GR mRNA and protein levels [61,62,63], which may result in the aforementioned reduced glucocorticoid feedback, together with an increased activity of both HPA and sympathetic-adrenal-medullary systems, as observed in prenatally undernourished subjects [57,64].

It is noteworthy that, in other studies, the GR mRNA expression in the PVN [57] and whole hypothalamus [65] of ewes, as well as the GR mRNA and protein in the rat hypothalamus [66], were not affected by previous exposure to prenatal undernutrition. Besides, Stevens et al. [67] have shown an increase in GR mRNA expression in the hypothalamus of perinatally undernourished ewes, while Li et al. [68] observed that PVN GR immunoreactivity was not affected in pre-term baboons (90% gestation) from mothers fed 70% of the control diet, although peptide expression and serum levels of ACTH and cortisol were found to be increased in the same animals. This apparently conflicting evidence could be the result of comparing GR levels between different animal species that may have different periods of vulnerability during intrauterine life, since ungulates (sheep) and primates (baboons) are born with more advanced neurobiological development (neurogenesis, synaptogenesis, gliogenesis, oligodendrocyte maturation, and age-dependent behaviors) than rats [67,68]. Furthermore, comparing data involving the PVN [57,68] vs. the entire hypothalamus [55,64,65,66], or changes in GR protein [56,68] vs. those in GR mRNA [57,65,67] may also lead to conflicting results.

Taken together, the aforementioned results suggest that increased CRF levels and subsequent hypertension caused by the activation of both the HPA axis and the sympathetic-adrenal-medullary network may occur in animals exposed to prenatal malnutrition also due to other independent intervening factors, unrelated to changes in GR expression in the PVN.

### 2.2. Increased Expression of CRF in the PVN and Subsequent Hypertension Shown by Prenatally Malnourished Animals Are Likely due to Central Noradrenergic Hyperactivity

One of the most likely factors involved in the increased expression of CRF in previously undernourished adult animals is the central noradrenergic hyperactivity that develops in these animals shortly after birth. Indeed, a body of literature has shown that the brains of prenatally malnourished rats exhibit increased synthesis, release, and turnover of noradrenaline [69,70,71,72,73,74,75,76,77,78,79]; for review see [7]. Extracellular noradrenaline levels were also recently reported to be significantly higher in both hemispheres, at all-time points, in adult animals that were malnourished during prenatal life compared to well-fed controls [80]. This finding confirmed older data [81], which found that extracellular noradrenaline levels in the prefrontal cortex were higher in perinatally protein-deprived rats compared to control ones, as measured by microdialysis.

The central noradrenergic hyperactivity found in the brain of malnourished animals has functional implications. Indeed, noradrenaline-dependent interhemispheric electrophysiological dominance is suppressed in these animals, an effect that may be prevented by chronic treatment with clonidine (a presynaptic adrenoreceptor agonist that could reduce noradrenaline release) [73] or with α-methyl-p-tyrosine (an inhibitor of tyrosine hydroxylase and thus of noradrenaline synthesis) [78]. Furthermore, the defective long-term potentiation (LTP) found in the prefrontal cortex of prenatally malnourished rats was restored to normal levels by antagonizing or generating knockdown of α_2C_-adrenoceptors, which suggests that the observed LTP neuroplastic deficit is generated by an excess of noradrenaline in the brain [82]. There are good reasons to consider central noradrenergic hyperactivity as a likely causal factor for both the increased levels of hypothalamic CRF and the downstream hypertensive effect developed by animals suffering from fetal malnutrition: (i) Axonal noradrenergic neurons from the A1, A2, A5, A6 (*locus coeruleus*) and A7 brainstem areas project densely to magnocellular and parvocellular regions of the PVN [83,84,85]; for in-depth review see [86]. (ii) In the PVN, noradrenaline stimulates transcription of the CRF gene very rapidly [87] by activating α_1_-adrenoceptors [88], with subsequent increase in cytosolic calcium in PVN neurons [89] and CRF release [90]. (iii) Microinjection of the α_1_-adrenoceptor agonist phenylephrine in the PVN excites parvocellular neurons [91] and increases blood pressure in eutrophic normotensive rats [92,93], while microinjection of the α_1_-adrenoceptor antagonist prazosin counteracts the hyperactivity of PVN neurons [52] and the hypertensive state observed in prenatally undernourished rats [24,52,93]. (iv) Electrical stimulation of the brainstem-PVN noradrenergic connection excites the majority of PVN neurons, the effects being counteracted by the α_1_-adrenoceptor antagonists ergotamine and prazosin and mimicked by the α_1_-adrenoceptor agonist phenylephrine [91,94,95,96,97]. In addition, elevated levels of basal neuronal activity were simultaneously found also at the locus coeruleus (LC) [52], suggesting that the hyperactivity of PVN neurons is secondary to hyperactive noradrenergic inputs from the LC to the PVN. Of note, both the firing rate and the number of spontaneously active cells in the LC had been previously described as significantly higher in perinatally undernourished rats than in controls [98]. Despite this last series of observations, it seems inappropriate to consider neuronal hyperactivity in the LC of animals exposed to prenatal malnutrition as the sole factor influencing CRF activity in the PVN—and thereby the downstream activation of both the HPA axis and the sympathetic-adrenal-medullary network—because neuronal activity in the LC is in turn influenced by many reciprocal neural inputs. For instance, in rats exposed to prenatal stress [99,100] or prenatal malnutrition [52] it has been reported that LC and PVN neurons interact reciprocally, as part of an excitatory LC-PVN closed-loop where the tonic neuronal activities of these nuclei mutually influence each other.

Thus, the above data strongly suggest that increased brainstem noradrenergic input to the NPV is at the origin of the CRF overexpression observed in early malnourished animals, which in turn leads to increased blood pressure through intensified neural and endocrine signaling. However, it is worth noting that while there are experimental data showing increased potassium-induced norepinephrine release in the cerebral cortex of early malnourished animals [72,73,76,79], which is released only from axons originating in the LC [101], similar determinations in the hypothalamus of malnourished animals are lacking. Indeed, while norepinephrine turnover is increased in the hypothalamus of early malnourished rats [75], hypothalamic norepinephrine release was not determined.

## 3. Maladaptive Programming of Paraventricular-Coerulear Network and Development of Hypertension in Prenatally Malnourished Adult Animals

In order for LC and PVN neurons to interact reciprocally as part of an excitatory closed loop between the LC and PVN in stressed and prenatally malnourished rats, where tonic neuronal activities in the two nuclei influence each other [52,93,99,100,102,103], in addition to the noradrenergic excitatory connection to the PVN there should be reciprocal excitatory pathways from the PVN to the LC. Available information in this regard indicates that such excitatory connections from the PVN to the LC are primarily provided by CRFergic innervation. Anatomical studies have shown nerve endings that are positive for CRF immunoreactivity, in apparent contact with neurons that exhibited tyrosine hydroxylase immunoreactivity and are therefore presumed to be noradrenergic [104,105]. Afterwards, unequivocal evidence was presented indicating that some peroxidase-labeled PVN neurons have monosynaptic associations with gold-silver labeled catecholaminergic dendrites in the LC [106], and that immunoreactive labeling of CRF receptors developed on perikarya and dendrites of tyrosine hydroxylase-positive neurons of the LC, as revealed by double labeling under epifluorescence [107] and electron microscopy [108]. Furthermore, many of the LC neurons can be excited by CRF microinjection into the nucleus [109,110] through activation of CRF_1_ receptors [111,112], eventually coexpressed with CRF_2_ receptors [113]. Of note, CRF microinjection into the LC of healthy normotensive rats, in addition to increasing the rate of neuronal activation in the LC also increased the neuronal activity in the PVN, an effect that was prevented by prior microinjection of the α_1_-adrenoceptor antagonist prazosin into the PVN. This evidence supports the existence of functional excitatory noradrenergic connections from the LC to the PVN triggered by the CRF microinjection into the LC [52]. Thus, a morphofunctional neural network—i.e., PVN-to-LC and LC-to-PVN excitatory connections—can be distinguished, in agreement with the proposed closed positive feedback loop that reciprocally interconnects the PVN and LC through CRFergic and noradrenergic projections.

### 3.1. The Paraventricular-Coerulear Network: Differences between Eutrophic and Prenatally Undernourished States

It is clear that in healthy normotensive animals (i) administration of norepinephrine or α_1_-adrenergic receptor agonists could activate PVN neurons [91,96,114], which results in hypertension and tachycardia [92,93], and (ii) CRF administration to the LC excites central LC neurons [109,110] through activation of CRF_1_ receptors [111,112], which also leads to increased blood pressure and heart rate [24,52,93,115]. These observations are consistent with the aforementioned anatomical and functional studies showing that some PVN and LC neurons are reciprocally and monosynaptically interconnected via CRFergic and noradrenergic axons. Interestingly, unlike the effects of agonists in the PVN and LC in normotensive eutrophic rats, the administration of the same agonists in the same brain sites in hypertensive undernourished animals did not produce any cardiovascular effect [24,93]. In contrast, administration of the α_1_-adrenoceptor antagonist prazosin into the PVN or the CRF receptor antagonist α-helical CRF into the LC, lowered blood pressure and heart rate in hypertensive undernourished animals only, failing to modify these cardiovascular parameters in normotensive eutrophic controls [24,93]. In summary, the antagonists of these receptors were active only in prenatally malnourished rats while the agonists were effective only in eutrophic animals, when microinjected into the aforementioned nuclei. Besides, both the CRF antagonist α-helical CRH [116] and the CRH_1_ selective antagonist antalarmin [117] were found to be antihypertensive when administered i.c.v. in acute rat models of hypertension, whereas i.c.v. administered α-helical CRF or injected near the LC did not alter the LC spontaneous neuronal firing rate in normotensive rats [118]. A similar picture emerges in animals under stress, since i.c.v. injection of the antagonist α-helical CRF significantly attenuated hypertension and tachycardia in stressed rats but not in unstressed subjects; in contrast, i.c.v. injection of the agonist, CRF, increased blood pressure and heart rate in unstressed rats [119]. Regarding α_1_-adrenoceptor ligands, intra-PVN microinjection of prazosin [120] or perfusion of the PVN with dialysate containing prazosin [121], showed that the α_1_-adrenoceptor antagonist did not alter control levels of mean arterial pressure or heart rate in normotensive rats. In contrast, intra-PVN microinjection of prazosin decreased the systolic pressure and heart rate in previously undernourished adult hypertensive rats [24,93] altogether with lowering the spontaneous firing rate in PVN neurons [52]. In contrast, both noradrenaline and the α_1_-adrenoceptor agonist phenylephrine increased the frequency of spontaneous excitatory postsynaptic currents in PVN slices from normotensive control animals, but not in PVN slices from animals made hypertensive by chronic intermittent hypoxia [122].

Why are agonists of α_1_-adrenoceptors and CRF receptors inactive in undernourished hypertensive animals? As stated elsewhere [93], the hypothesis of agonist inefficacy in undernourished animals caused by desensitization of α_1_-adrenergic and CRFergic receptors due to hyperactivity exhibited by central noradrenergic and CRFergic systems is not tenable, because the respective antagonists are in fact active at these receptors in the same animal models. Instead, because in undernourished rats the LC [52,98] and PVN [52] basal neuronal activity is about twice that of normal rats, it was argued that in these animals the PVN and LC neurons are already fully active and therefore insensitive to further excitation by application of exogenous agonists [52]. In fact, the spontaneous discharge rate of LC neurons may not increase more than two-fold after high doses of both intra-LC [109] and i.c.v. CRF [109,123], which appears to work as a frequency limit for the rate of neuronal firing, at least in these two nuclei. A similar result was found in rats chronically stressed by separation from the mother during lactation, where the LC neurons had spontaneous firing rates two-fold higher than those of controls [110]. In those stressed rats, CRF application did not further activate LC neurons but did increase LC firing rate in LC neurons in control rats [110]. Regarding the increased spontaneous neuronal rhythm found in the LC of previously malnourished animals, it has been proposed that this might be the consequence of reduced negative feedback mediated by somatodendritic α_2_ autoreceptors, which are found to be decreased in the LC of perinatally undernourished rats [98]. However, later studies showed that α_2_-adrenoceptors are increased, at least in the cerebral cortex of adult previously undernourished rats, as detected by [^3^H]-rauwolscine binding [124]. Therefore, the fact that LC and PVN neurons in undernourished animals have a higher rate of spontaneous firing remains yet unexplained (but see below).

Another intriguing issue is why α_1_-adrenoceptor and CRF receptor antagonists, unlike agonists, do not modify blood pressure and heart rate in normotensive eutrophic animals. Prazosin is a selective inverse agonist for α_1_-adrenoceptors [125] that binds almost equally the α_1A_, α_1B_, and α_1D_ adrenoceptor subtypes [126], while α-helical CRF is a non-selective competitive antagonist for CRF_1_ and the splice variants CRF_2(a)_ and CRF_2(b)_ receptors in the rat [127]. As competitive ligands, their ability to induce receptor-mediated effects requires prior receptor activation by tonically released noradrenaline in the PVN and CRF in the LC. Thus, the fact that microinjection into the PVN and LC of the aforementioned ligands did not produce hypotension, might lead to the notion that in healthy normotensive rats there is no tonic release of norepinephrine and CRF in the PVN and CL, respectively, or that such tonic neurotransmitters release is not, significantly involved in normal basal values of blood pressure. Therefore, it seems apparent that an excitatory noradrenergic/CRFergic feed-forward loop interconnecting reciprocally the PVN with the LC, which necessarily needs tonic activity in at least one of these two nuclei, is enabled in undernourished animals by some type of neuroplastic adaptative mechanism. Interruption of one of the two arms in the reciprocal communication between PVN and LC, both in malnourished hypertensive and eutrophic normotensive animals, could shed some light on this mechanism.

### 3.2. Disruption of the PVN-LC Reciprocal Communication: Effects on Neuronal Activity and Cardiovascular Parameters in Undernourished and Eutrophic Animals

It is noteworthy that the transient cardiovascular effects of agonists observed in healthy normotensive animals, i.e., intra-PVN phenylephrine or intra-LC CRF, were not prevented by disruption of the reciprocal communication between the PVN and the LC using appropriate antagonists, i.e., α-helical CRF intra LC [93] or prazosin intra-PVN [24,93]. This means that the hypertension and increased heart rate seen in healthy normotensive rats by agonist-induced excitation of either PVN or LC neurons did not involve paraventricular-coerulear excitatory interactions. In other words, the excitation of PVN or LC neurons by the respective agonist can independently activate the sympathetic–adrenal–medullary system and thus generate the described cardiovascular effects, without the need for coactivation of the complementary nucleus. In contrast, disruptions in malnourished rats of the CRFergic connection from the PVN to the LC with intra-LC α-helical CRF, or the reciprocal noradrenergic connection from the LC to the PVN with intra-PVN prazosin, were both independently capable of suppressing the hypertension and bradycardia in these animals [93]. This implies that a feed-forward closed loop of mutual excitation between both nuclei is required for producing the cardiovascular effects. Indeed, intra-PVN microinjection of prazosin in prenatally malnourished rats has been found to depress the neuronal firing that is increased in the PVN of those animals, but also in the LC, probably due to the removal of the CRFergic arm of the PVN-LC bidirectional communication pathway [52]. This would interrupt the feed-forward loop of mutual excitation between both nuclei, thus producing a drop in blood pressure and heart rate in the hypertensive undernourished animals by decreasing the output of the sympathetic-adrenal-medullary system. Although disruption of the reciprocal α_1_-adrenergic arm of the closed loop also reduced blood pressure and heart rate in malnourished rats [93], the effect of CRF receptor blockade in LC neurons of those animals has yet to be tested electrophysiologically. Finally, administration of an antagonist in one of the nuclei (i.e., prazosin in the PVN) in undernourished rats effectively allowed the complementary nucleus (i.e., the LC) to recover full responsiveness to the agonist administered (in this case, CRF), now inducing hypertension and tachycardia [93]. This means that reduction in tonic activity in the LC by suppressing the PVN-LC CRFergic communication is enough to rescue the ability of LC neurons to respond to the agonist in these animals [93]. Similar cardiovascular responses are produced when the antagonist (α-helical CRF) is administered to the LC and the agonist (phenylephrine) to the PVN [93]. This issue strongly supports the contention that agonists are inactive in undernourished animals because LC and PVN neurons are already fully active and therefore insensitive to further excitation by application of exogenous agonists.

Taken together, the above data indicate that a reciprocal excitatory feedforward loop between CRFergic neurons of the PVN and noradrenergic neurons of the LC is enabled in animals malnourished during fetal life, which would lead to increased tonic activity of these neurons, while this loop is functionally absent in the eutrophic control counterpart, or at least not working properly (see Figure 1 for summary of supporting experimental protocols). This permissive mechanism could well be at the base of the high spontaneous firing rate of neurons in the LC and PVN from undernourished animals, escalating beyond the firing rate values found in the eutrophic animals, with the consecutive hypertension and tachycardia.

Importantly, the reciprocal excitatory feedback loop between PVN and LC neurons in prenatally malnourished adult rats is certainly more complex when considering the local regulatory neuronal circuits in these nuclei, which include some well-characterized modulatory interneurons. In fact, in the PVN, together with the direct α_1_-adrenoceptor-mediated excitatory noradrenergic input to CRF-expressing parvocellular neurons [128,129,130,131,132], there are GABA-synthesizing interneurons that inhibit PVN neuronal activity [133]. The noradrenergic input to the PVN has been shown to inhibit PVN-surrounding GABAergic interneurons via α_2_-adrenoceptor activation, resulting in disinhibition of the parvocellular PVN neurons [91,134] altogether with an increased sympathetic activity triggered from the PVN [135], thus reinforcing a α_1_-adrenoceptor-mediated direct activating effect on parvocellular neurons of the PVN. In addition, noradrenergic input also activates local intra-PVN glutamatergic interneurons via α_1_-adrenoceptors [136], leading to indirect parallel heterosynaptic activation of parvocellular PVN neurons, increasing once again the hypertensive effects and tachycardia. With respect to the modulatory influences on the LC, a rather similar picture emerges because in addition to the excitatory CRF input coming into LC norepinephrine-synthesizing neurons [137,138,139] these neurons also receive synaptic contacts from GABAergic peri-LC interneurons that provide inhibitory regulation mediated by GABA_A_ receptors [140,141]. Anatomical, electrophysiological and optogenetic data showed that noradrenergic LC-core neurons and GABAergic peri-LC neurons receive axon afferents from different brain regions concerned with processing and evaluation of sensory stimuli and stressors. However, the input coming from the PVN arrives rather directly to noradrenaline-expressing LC-core neurons [141], thus supporting a direct PVN-LC connection as part of the feed-forward excitatory loop that could be enabled in prenatally undernourished adult rats (see Figure 2).

Which are the signaling mechanisms to the reciprocal excitatory PVN-LC feedforward loop in animals undernourished early in life? The underlying neural modifications that may underpin the recruitment of such a feedforward closed loop are likely based on epigenetic adaptive molecular changes involved in controlling the excitability and neuroplasticity of specific subsets of noradrenergic and CRFergic long-axon neurons, together with the GABAergic and glutamatergic interneurons existing in the PVN and the LC. In this regard, while no reports on epigenetic modifications induced by early undernutrition affecting noradrenergic and/or CRFergic systems that may be related to hypertension exist to date, it is known that human undernutrition is normally accompanied by other nutritional deficiencies (i.e., folate and polyunsaturated fatty acids), which can lead to epigenetic changes, for instance, in GABA systems of the brain. Indeed, folate deficiency during development results in epigenetically increased homocysteine levels, a molecule that competes with GABA for GABAergic receptors thus reducing the effects of GABA in the brain [142], while deficits of polyunsaturated fatty acids may lead to hypermethylation of some gene promoters resulting in downregulated expression of GABA-related genes [143]. Thus, dietary deficits of both folate and polyunsaturated fatty acids could theoretically program epigenetic changes in brain GABA-related circuits that may lead to neuronal disinhibition. However, it must be taken into account that protein-deficient purified diets used to study the effects of malnutrition in experimental animals are usually compensated for with an excess of folate and sometimes with polyunsaturated fatty acids, which clearly precludes comparison with human protein/energy malnutrition.

## 4. PVN as the Output System That Mediates Hypertension and Increased Heart Rate during Tonic Activation of the Paraventricular-Coerulear Network in Prenatally Malnourished Animals

The natural question that arises from the aforementioned studies is: are the PVN and/or the LC the output point of the excitatory feedforward PVN-LC closed loop installed in undernourished animals? It is well established that the PVN provides a dominant source of excitatory drive to the cardiovascular system via a sympathetic outflow [54,144,145]. Indeed, a critical CRFergic pathway that originates in parvocellular cells of the PVN is projected to the rostral ventrolateral medulla [106,146]. These CRFergic neurons, in turn, have a direct and highly potent synaptic relationship with spinal preganglionic sympathetic neurons that control the sympathetic output to different target organs involved in the regulation of blood pressure, including the heart and blood vessels, the kidney, and the adrenal medulla. As already mentioned, PVN neurons are usually activated by α_1_-adrenoceptor agonists such as phenylephrine [83,91,128], and α_1_-adrenoceptor stimulation with phenylephrine in the PVN has been found to induce cardiac chronotropic and inotropic responses [92]. In contrast, the LC is involved in the integration and distribution of stress-related afferent signals to forebrain structures like the cerebral cortex, hippocampus, cerebellum, most thalamic nuclei, and partially the hypothalamus. In addition, LC neurons, which are activated by CRH [109] via CRF_1_ receptors [111,112], could also modulate blood pressure and heart rate, as LC projects to sympathetic preganglionic spinal neurons, which are in turn excited by the noradrenaline released through activation of α_1_-adrenoceptors [101]. However, LC stimulation is associated with moderate and variable increases in heart rate and blood pressure [147] because LC also projects to the rostral ventrolateral medulla [148] where noradrenaline exerts an inhibitory effect via stimulation of α_2_-adrenoceptors [149], thus dampening the sympathoexcitation evoked in preganglionic spinal neurons [101]. 

It seems very likely that the cardiovascular effects generated by intra-LC microinjected CRF are exerted primarily by transferring neuronal excitation from LC to PVN neurons via the α_1_-adrenoceptor-mediated excitatory pathways mentioned above, with PVN being the output system responsible for the increases in systolic pressure and the heart rate. This view is strongly supported by data showing that intra-LC CRF-induced hypertension and tachycardia in normotensive healthy rats are both suppressed by blocking α_1_-adrenoceptors in the PVN with prazosin [24,52,93]. A similar reasoning could also apply in fact to the hypertensive state associated to prenatal undernutrition and maintained by the PVN-LC excitatory feedforward closed loop described, where the PVN output would be translated to sympathoexcitation. Indeed, it seems clear today that development of hypertension in rats exposed to protein restriction during pregnancy and lactation is associated with sympathetic hyperactivity [150]. It should be noted, however, that today it is apparent that PVN neurons do not appear to play an essential role in regulating the sympathetic response to short-term cardiovascular changes. Rather, they seem to be involved in long-term challenges such as sustained water deprivation, chronic hypoxia, pregnancy, stress, and other forms of enduring hypertension [144]. 

The establishment of the PVN nucleus as a main output center that transfers increased neuronal activity to elevated blood pressure and tachycardia requires at least two main conditions: The PVN should not have a negative feedback mechanism associated with the down-regulation of PVN neurons (i.e., some inhibitory effect exerted backward on α_1_-adrenoceptor expression on PVN neurons). The absence of inhibition is required to sustain the continuous overactivity of the PVN in a pathological condition, such as in prenatal undernutrition. The evidence shows that prenatal undernutrition did not change the amounts of α_1_-adrenoceptor binding sites in the hypothalamus determined with [^3^H]-prazosin, but a lower expression of α_1A_-adrenoceptor mRNA was measured by in situ hybridization [93]. In that study [^3^H]-prazosin binding identified all three α_1_-adrenoceptor subtypes [126], while the deoxynucleotide probe used was specific for the α_1A_-adrenoceptor mRNA subtype [93]. It is also noteworthy that the binding assay was performed in the entire hypothalamus, while in-situ hybridization allowed specific recognition of mRNA in delimited regions of the PVN [93]. In this context, further experiments are needed to establish the net effect on the whole α_1_-adrenoceptor spectrum. This issue is of paramount importance because the three α_1_-adrenoceptor subtypes are expressed in the PVN [151,152,153], and they suffer different regulation processes depending on the primary condition. For example, α_1D_ up-regulates while the α_1A_ and α_1B_ subtypes down-regulate in a concentration-dependent manner during an agonist challenge (i.e., endogenous noradrenaline), and down-regulation of the latter was accompanied by reductions of mRNA (for review see [154]). Interestingly, higher levels of α_1A_-adrenoceptor mRNA have reported in the PVN of rats suffering from chronic hypertension [155], but unchanged α_1A_, α_1B_, or α_1D_-adrenoceptor mRNA in the whole hypothalamus [156] and the VPN [157] has also been reported. Besides, chronic stress (which is often accompanied by hypertension) sensitizes the HPA axis to further acute stress (as measured by transient plasma ACTH increase) in rats, enhancing the response to α_1_-adrenergic receptor activation in the PVN [158]. Thus, despite the implicit importance of the above results, the specific regulation mechanisms of α_1_-adrenoceptor subtypes due to prenatal nutritional maladaptive programming remains still unknown.Permanent sensitization of the PVN should be required to maintain the integrity of the excitatory feed-forward loop. In this regard, it has been reported that noradrenaline induces an α_1_-adrenoceptor-mediated increase and an α_2_-adrenoceptor-mediated decrease in GABA-dependent spontaneous inhibitory postsynaptic current in a subset of parvocellular neurons of the PVN [159], which possibly represents a metaplastic regulation of GABAergic transmission in these neurons. Hippocampal long-term potentiation [160] and cerebral cortex long-term depression [161,162] have been reported also to be promoted by α_1_-adrenoceptors, but studies on neuroplasticity processes involved in long-lasting sensitization of neurons in the PVN, which can promote an enduring sympathetic activation thus favoring chronic hypertension, are still lacking.

## 5. Conclusions

Taken together, the evidence reviewed may be summarized as follows: Both hypertension and tachycardia induced in healthy normotensive rats by either α_1_-adrenoceptor-mediated excitation of PVN neurons or CRF receptor-mediated excitation of LC neurons do not imply serial or reciprocal excitatory interactions between the two nuclei, as revealed by the fact that the cardiovascular effects observed were not prevented by disruption of the communication between the nuclei.Simultaneous concurrent tonic neuronal activity in the PVN and the LC is required to maintain elevated arterial blood pressure and heart rate scores in prenatally malnourished animals. In addition, reciprocal noradrenergic and CRFergic excitatory connections between the PVN and the LC give rise to a feedforward paraventricular-coerulear closed loop of neuronal activity, which is an essential part of the molecular etiological component of hypertension and tachycardia generated in animals submitted to prenatal undernutrition.The PVN may act as the exit point of the paraventricular-coerulear loop that downstream activates the sympathetic system, producing hypertension and tachycardia in malnourished animals. As such, it is essential that α_1_-adrenoceptor desensitization does not occur in the PVN of malnourished rats, allowing the PVN to function as the output locus in the paraventricular-coerulear network. More research is required to support this point.Whether noradrenergic hyperactivity in prenatally and perinatally undernourished animals is the primary factor involved in the triggering of neuronal hyperactivity in the PVN-LC communication, or on the contrary, it is a consequence of activity in such an interconnected set of neurons, is not entirely clear at present. Additionally, whether some epigenetic mechanisms may be underlying some of the remarkable characteristics that such an interactive neural system acquires under conditions of malnutrition remains unknown. Indeed, both early-life stress and early-life undernutrition similarly led to life-long alterations in the neuroendocrine stress system, partially by modifying epigenetic regulation of gene expression [163]. Increased CRF production via epigenetic mechanisms cannot be discarded since prenatal restraint stress is associated with the demethylation of CRF promoter, thereby enhancing CRF transcriptional responses to stress in adolescent rats [164]. However, no epigenetic modifications underlying altered CRF expression in prenatally undernourished animals have been reported so far.Other central nervous system programming factors that may underlie hypertension due to prenatal undernutrition, such as enhanced sympathetic-respiratory coupling at early life, inappropriate activation of the renin-angiotensin system, glucocorticoid neuronal remodeling, should not be neglected. Carefully designed experimental protocols should be arranged in order to study the specific contributions of those neural/endocrine components as well as the possibility of a relationship with the neuronal hyperactivity in the paraventricular-coerulear network.

As a final consideration, it is worth to highlight that further investigation is required to generate new data that may describe the mechanisms involved in each of these relevant aspects, which may shed light on the functional link between malnutrition and pathological programming of hypertension in humans.

## Figures and Tables

**Figure 1 ijms-23-11965-f001:**
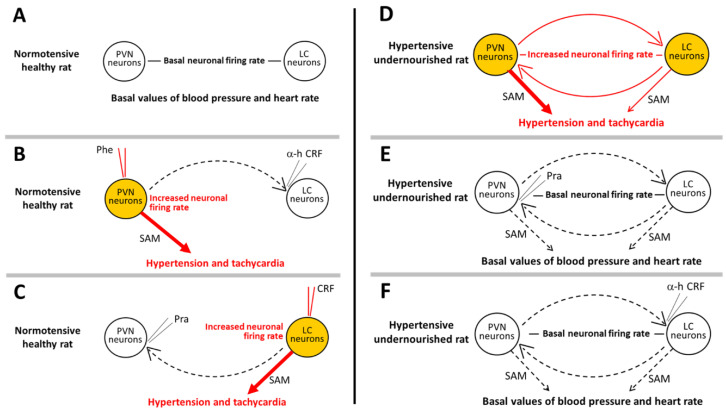
Reciprocal excitatory feedforward loop between CRFergic neurons of the PVN and noradrenergic neurons of the LC, enabled in prenatally malnourished animals. **Left panels** (**A**–**C**) refer to the level of neuronal activity in PVN and LC as well as the levels of blood pressure and heart rate, in healthy normotensive rats: (**A**) at rest with no drugs (white color means basal neuronal activity in PVN and LC, along with normal basal blood pressure and heart rate), (**B**) after stimulation of PVN neurons with the α_1_-adrenoceptor agonist phenylephrine while blocking the PVN-LC communication with the CRF receptor antagonist α-helical CRF (orange color at PVN means increased neuronal activity and hypertension/tachycardia), (**C**) after stimulation of LC neurons with CRF while blocking the LC-PVN communication with the α_1_-adrenoceptor antagonist prazosin (orange color at LC means increased neuronal activity and hypertension/tachycardia). **Right panels** (**D**–**F**) refer to the level of neuronal activity in PVN and LC as well as the levels of blood pressure and heart rate, in undernourished hypertensive rats: (**D**) with no drugs (orange color at PVN and LC means increased neuronal activity in both nuclei, along with hypertension and tachycardia), (**E**) after blocking the LC-PVN communication at the PVN with the α_1_-adrenoceptor antagonist prazosin (white color at PVN and LC means basal neuronal activity in both nuclei, along with normal blood pressure), (**F**) after blocking the PVN-LC communication at the LC with the CRF receptor antagonist α-helical CRF (white color at PVN and LC means basal neuronal activity in both nuclei, along with normal blood pressure). PVN, paraventricular nucleus of the hypothalamus; LC, locus coeruleus; SAM, sympathetic-adrenal-medullary output system (thick arrows indicate larger output). Continuous red curved arrows indicate enabled communication between nuclei, and segmented black curved arrows indicate disrupted communication. Note red micropipettes for injecting the agonists, and grey micropipettes for injecting the antagonists. Overall, agonists produce increased neuronal firing rate in normotensive healthy animals leading to hypertension and tachycardia, the antagonists in the complementary nuclei preventing the excitation there and the cardiovascular effects. In hypertensive undernourished animals, any of the two antagonists turn off the increased neuronal firing rate showing these animals both in the PVN and LC, thus normalizing the blood pressure and heart rate.

**Figure 2 ijms-23-11965-f002:**
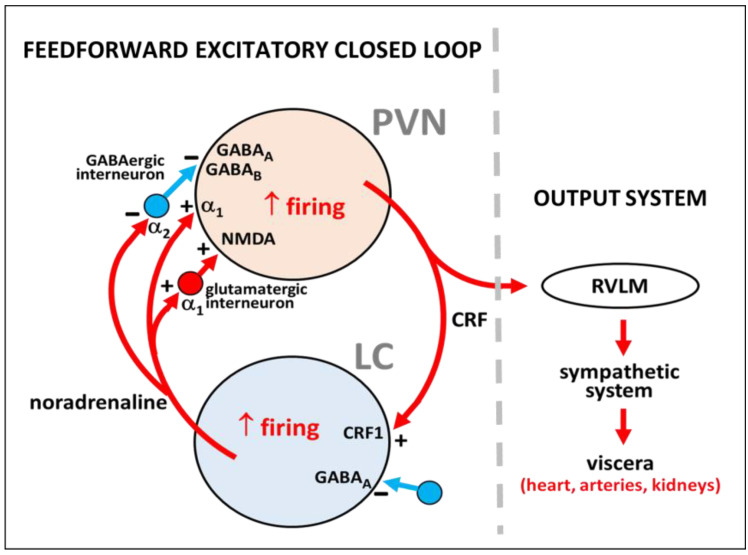
Summary of local regulatory neural circuits in the PVN-LC network considering the role of interneurons in the PVN and the LC according to the literature. A CRF-expressing neuron of the PVN and a noradrenaline-synthesizing neuron of the LC are represented, showing α_1_ and α_2_ receptors for noradrenaline, GABA_A_ and GABA_B_ receptors for γ-aminobutyric acid, NMDA receptor for glutamate, and CRF_1_ receptor for CRF. Excitation or inhibition are indicated by the plus (+) and minus (−) signs, respectively. The output system to viscera is shown. RVLM, rostral ventrolateral medulla.

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
