# Peer review of "The Role of the Paraventricular-Coerulear Network on the Programming of Hypertension by Prenatal Undernutrition"

_ijms, 2022, doi:10.3390/ijms231911965_

Round 1

Reviewer 1 Report

The authors reviewed the role of the  paraventricular nucleus and locus coeruleus in hypertension induced by prenatal undernutrition. Considering available literature there is a space for such a review. 

1) The paper contains useful information concerning role of the central nervous system. However, the authors should shortly discus the role of brain in the context of potential peripheral mechanisms. Such integration would help readers and would increase an impact of the paper.

2) Please provide information about percentage of humans that experienced prenatal undernutrition and later developed hypertension.

3) In the sentence “In turn, ACTH induces the synthesis of glucocorticoids….’” please add “ and release of”.

4)The entire paragraph (lines 95-104) does not contain any citations. Please, provide some references for this text.

5)The paragraph 161-171 have improper logical structure. The word “moreover” implies that the sentence provides additional information supporting the previous sentence. However, in the manuscript the statement following “moreover” contains contradictory information (have shown an increase in GR mRNA expression in the hypothalamus of perinatally undernourished ewes). Please, rewrite this part. Furthermore, do you have some ideas explaining inconsistent findings?

6) The sentence “…there must be reciprocal excitatory pathways between  these nuclei despite of the unidirectional LC – PVN noradrenergic excitatory connection” is strange. Obviously, noradrenergic connection is unidirectional because noradrenaline is released only by very restricted number of cell localized in LC. However, there is no logical conflict between presence of unidirectional noradrenergic connection and reciprocal connection mediated by other neurotransmitters.

7) The sentence “Taken together, the above data indicate that a reciprocal excitatory feedforward loop between CRFergic neurons of the PVN and noradrenergic neurons of the LC is enabled in animals malnourished during fetal life, which would lead to increased tonic activity of these neurons, while this loop is absent in the eutrophic control counterpart” is not clear enough. The information “this loop is absent” suggests structural differences in brain wiring although I think that the authors mean that the loop is functionally different. Please, specify.

Author Response

Point by point Responses to the Reviewers’ comments

We thank the reviewers for their constructive comments on our manuscript.

Lines in parentheses refer to the new manuscript.

REVIEWER 1.

1) The paper contains useful information concerning role of the central nervous system. However, the authors should shortly discus the role of brain in the context of potential peripheral mechanisms. Such integration would help readers and would increase an impact of the paper.

Response: The idea of brain-driven peripheral mechanisms is explicitly stated in several places throughout the manuscript (including the new Figure 2). We understand the reviewer's concern, but we found it difficult to fit a specific paragraph with that goal in a certain place in the manuscript.

2) Please provide information about percentage of humans that experienced prenatal undernutrition and later developed hypertension.

Response: A new paragraph dealing with humans experiencing prenatal undernutrition and later developing hypertension was added (lines 69 to 82).

3) In the sentence “In turn, ACTH induces the synthesis of glucocorticoids….’” please add “ and release of”.

Response: Done (line 122).

4)The entire paragraph (lines 95-104) does not contain any citations. Please, provide some references for this text.

Response: Four references were added on the specified text (references 47 to 50).

5)The paragraph 161-171 have improper logical structure. The word “moreover” implies that the sentence provides additional information supporting the previous sentence. However, in the manuscript the statement following “moreover” contains contradictory information (have shown an increase in GR mRNA expression in the hypothalamus of perinatally undernourished ewes). Please, rewrite this part. Furthermore, do you have some ideas explaining inconsistent findings?

Response: The paragraph was re-written and possible reasons for contradictory information provided (lines 183-197).

6) The sentence “…there must be reciprocal excitatory pathways between these nuclei despite of the unidirectional LC – PVN noradrenergic excitatory connection” is strange. Obviously, noradrenergic connection is unidirectional because noradrenaline is released only by very restricted number of cells localized in LC. However, there is no logical conflict between presence of unidirectional noradrenergic connection and reciprocal connection mediated by other neurotransmitters.

Response: The paragraph was re-written (lines 262-266).

7) The sentence “Taken together, the above data indicate that a reciprocal excitatory feedforward loop between CRFergic neurons of the PVN and noradrenergic neurons of the LC is enabled in animals malnourished during fetal life, which would lead to increased tonic activity of these neurons, while this loop is absent in the eutrophic control counterpart” is not clear enough. The information “this loop is absent” suggests structural differences in brain wiring although I think that the authors mean that the loop is functionally different. Please, specify.

Response: The paragraph was re-written (lines 396-400).

Reviewer 2 Report

General Concerns:

In this descriptive review Cayupe et al.  summarized the  neuroendocrine and neuroplastic modifications that underlie maladaptive mechanisms related to  chronic hypertension programming after early undernutrition. This manuscript need some of refinement:

i) The authors should explain what mean caloric and protein undernutrition in term of calories value? And in protein-deficient term?Thus, the review should describe low birth weight in term of BMI value?

ii) From line 95 to 104 missing significant references.

iii) The conclusions are very expansive and dispersive.  

Minor:

In line 76 what mean in SHR animals, explain to the reader please?

Author Response

Point by point Responses to the Reviewers’ comments

We thank the reviewers for the constructive comments on our manuscript.

Lines in parentheses refer to the new manuscript.

REVIEWER 2.

In this descriptive review, Cayupe et al. summarized the neuroendocrine and neuroplastic modifications that underlie maladaptive mechanisms related to chronic hypertension programming after early undernutrition. This manuscript need some of refinement:

  1. i) The authors should explain what mean caloric and protein undernutrition in term of calories value? And in protein-deficient term? Thus, the review should describe low birth weight in term of BMI value?

Response: Done. Explanations on this were added as a new paragraph (lines 47 to 56).

  1. ii) From line 95 to 104 missing significant references.

Response: Four references were added on the specified text (references 47 to 50).

iii) The conclusions are very expansive and dispersive. 

Response: The conclusions were revised. They also included areas where supporting information is yet missing, which may be contributing to the 'scattered' appearance of the conclusions.

  1. iv) In line 76 what mean in SHR animals, explain to the reader please?

Response: SRH replaced by “spontaneously hypertensive rats” (line 98).

Reviewer 3 Report

This is a comprehensive narrative review of data provided by animal and human studies on the central neuroendocrine and neuroplastic mechanisms underlying the programming of fetal malnutrition-induced hypertension. The data provided in this manuscript are of interest to researchers and clinicians.

 Minor queries:

-        The abbreviation SHR (line 76) should be replaced by the full term, as it appears only once in the text.

-        In Conclusions I suggest stating (line 554) “…programming of hypertension in humans”, but I leave it to the authors' consideration.

Author Response

Point by Point Responses to the Reviewers’ comments

We thank the reviewers for the constructive comments on our manuscript.

Lines in parentheses refer to the new manuscript.

REVIEWER 3.

This is a comprehensive narrative review of data provided by animal and human studies on the central neuroendocrine and neuroplastic mechanisms underlying the programming of fetal malnutrition-induced hypertension. The data provided in this manuscript are of interest to researchers and clinicians.

Minor queries:

- The abbreviation SHR (line 76) should be replaced by the full term, as it appears only once in the text.

Response: SRH replaced by “spontaneously hypertensive rats” (line 98).

- In Conclusions I suggest stating (line 554) “…programming of hypertension in humans”, but I leave it to the authors' consideration.

Response: Done (line 570).

Reviewer 4 Report

This review is based on the previous work of the authors in the field. Particular attention is given to the results published in one paper in 2021 repeating perhaps too many details.

Comments

  1. The authors should check the abbreviations. For example, the abbreviation for the HPA as well as ACTH was introduced twice.

  1. Most of the paragraphs are very long and the orientation in the text is difficult.

  1. In my opinion, it is not optimal to include a figure that has already been published in a recent open access publication. Instead, the authors might like to include an original figure(s) in other parts of the review to make the text more attractive for readers.

4. Repeating the ideas already mentioned in the paper of the authors published in 2021 should be reduced.

Author Response

Point by Point Responses to the Reviewers’ comments

We thank the reviewers for the constructive comments on our manuscript.

Lines in parentheses refer to the new manuscript.

REVIEWER 4.

This review is based on the previous work of the authors in the field. Particular attention is given to the results published in one paper in 2021 repeating perhaps too many details.

  1. The authors should check the abbreviations. For example, the abbreviation for the HPA as well as ACTH was introduced twice.

Response: Corrected.

  1. Most of the paragraphs are very long and the orientation in the text is difficult.

Response: We tried to correct this aspect as much as we could. Each long paragraph was divided into two shorter ones separated by a period (e.g., lines 285, 317, 369, 378, 428, 448, 563).

  1. In my opinion, it is not optimal to include a figure that has already been published in a recent open access publication. Instead, the authors might like to include an original figure(s) in other parts of the review to make the text more attractive for readers.

Response: Figure 2 was replaced by a new Figure 2 (not published before).

  1. Repeating the ideas already mentioned in the paper of the authors published in 2021 should be reduced.

Response: With respect to the ideas expressed in another document of ours, we retained only those necessary and sufficient for a complete discussion of the subject.

Round 2

Reviewer 2 Report

Accept in present form.

Reviewer 4 Report

The authors respended adequately to my comments.